# Partially Alternative Feeding with Fermented Distillers’ Grains Modulates Gastrointestinal Flora and Metabolic Profile in Guanling Cattle

**DOI:** 10.3390/ani13223437

**Published:** 2023-11-07

**Authors:** Guangxia He, Chao Chen, Shihui Mei, Ze Chen, Rong Zhang, Tiantian Zhang, Duhan Xu, Mingming Zhu, Xiaofen Luo, Chengrong Zeng, Bijun Zhou, Kaigong Wang, Erpeng Zhu, Zhentao Cheng

**Affiliations:** 1College of Animal Science, Guizhou University, Guiyang 550025, China; guangxiahe0@163.com (G.H.); chenc@gzu.edu.cn (C.C.); shihuimei0@163.com (S.M.); 14785135914@163.com (Z.C.); zr15185362945@163.com (R.Z.); ztt18286158817@163.com (T.Z.); l8718227x1@163.com (D.X.); 18608518394@163.com (M.Z.); 13368545574@163.com (X.L.); 18208427639@139.com (C.Z.); bjzhou@gzu.edu.cn (B.Z.); as.kgwang@gzu.edu.cn (K.W.); 2Guizhou Provincial Animal Disease Research Laboratory, Guiyang 550025, China

**Keywords:** fermented distillers’ grains (FDG), gastrointestinal microbiota, metabolic pathways, Guanling cattle

## Abstract

**Simple Summary:**

Fermented distillers’ grains are a common dietary supplement for livestock and poultry, aimed at improving health and metabolism. Despite this, there is a lack of studies examining the effect of fermented distillers’ grains intake on the gastrointestinal microbiota and metabolites in cattle. The objective of this study was to examine the effects of partially substituting traditional feed with fermented distillers’ grains on the gastrointestinal flora and metabolic profile of Guanling cattle. The results indicate that the fermented distillers’ grains replacement diet altered the microbial community structure in the abomasum and cecum of the cattle. Furthermore, it increased the abundance of probiotics while decreasing the abundance of potential pathogenic bacteria, thereby improving the overall gastrointestinal health of the cattle to some extent. The differential metabolites present in the abomasum and cecum contents of Guanling cattle demonstrated a significant enrichment in metabolic pathways, including primary bile acid biosynthesis and choline metabolism in cancer. Moreover, a noteworthy correlation was observed between the differential metabolites and the differential bacterial genera. This study’s outcomes offer a scientific reference for the potential employment of FDG as a feed resource in cattle.

**Abstract:**

Fermented distillers’ grains (FDG) are commonly used to enhance the health and metabolic processes of livestock and poultry by regulating the composition and activity of the intestinal microbiota. Nevertheless, there is a scarcity of research on the effects of the FDG diet on the gastrointestinal microbiota and its metabolites in cattle. This study examines the impact of FDG dietary supplements on the gastrointestinal flora and metabolic profile of Guanling cattle. Eighteen cattle were randomly assigned to three treatment groups with six replicates per group. The treatments included a basal diet (BD), a 15% concentrate replaced by FDG (15% FDG) in the basal diet, and a 30% concentrate replaced by FDG (30% FDG) in the basal diet. Each group was fed for a duration of 60 days. At the conclusion of the experimental period, three cattle were randomly chosen from each group for slaughter and the microbial community structure and metabolic mapping of their abomasal and cecal contents were analyzed, utilizing 16S rDNA sequencing and LC-MS technology, respectively. At the phylum level, there was a significant increase in Bacteroidetes in both the abomasum and cecum for the 30%FDG group (*p* < 0.05). Additionally, there was a significant reduction in potential pathogenic bacteria such as Spirochetes and Proteobacteria for both the 15%FDG and 30%FDG groups (*p* < 0.05). At the genus level, there was a significant increase (*p* < 0.05) in Ruminococcaceae_UCG-010, Prevotellaceae_UCG-001, and Ruminococcaceae_UCG-005 fiber degradation bacteria. Non-target metabolomics analysis indicated that the FDG diet significantly impacted primary bile acid biosynthesis, bile secretion, choline metabolism in cancer, and other metabolic pathways (*p* < 0.05). There is a noteworthy correlation between the diverse bacterial genera and metabolites found in the abomasal and cecal contents of Guanling cattle, as demonstrated by correlation analysis. In conclusion, our findings suggest that partially substituting FDG for conventional feed leads to beneficial effects on both the structure of the gastrointestinal microbial community and the metabolism of its contents in Guanling cattle. These findings offer a scientific point of reference for the further use of FDG as a cattle feed resource.

## 1. Introduction

With the rapid development of the beef cattle industry, the shortage of feed is becoming an urgent problem, and the high cost of breeding hinders the sustainable development of animal husbandry, which makes it necessary to seek new feed ingredients [1]. Distillers’ grains (DGs) are solid by-products produced after the processing and production of liquor [2], and the high content of water and organic matter makes them susceptible to rot and thus results in surrounding environmental pollution [3]. Generally, DGs are mainly used as feed for ruminants after simple drying and even without pretreatment. Due to the high content of rice husks, DGs are hard and rough, which makes them difficult to digest by livestock. Studies have also shown that DGs contain fat, protein, cellulose, vitamins, trace elements, and many other nutrients that are not fully utilized [4], which causes a great waste of resources. In recent years, researchers have used microbial (especially probiotics) fermentation to improve the nutritional value and stability, and resolve storage problems of DGs [5]. Meanwhile, fermented distillers’ grains (FDG) can not only be used as a feed resource to make up for the shortage of protein feed raw materials, which has prospective application and economic value in animal husbandry, but can also obviously alleviate environmental pollution problems caused by the improper disposal of wine lees, which is of great social significance.

The use of FDG as feed has been well studied in recent years. Research has shown that dietary fermented distillers’ grains can regulate growth performance and gut microbiota, and improve the carcass shape and meat quality of finishing pigs [6]. Feeding soluble distillers’ grains did not negatively affect the growth performance, carcass characteristics, serum parameters, and intestinal morphology of Cherry Valley ducks [7], and in particular, feeding fermented distillers’ grains also increased their final body weight. The partial replacement of soybean meal with fermented distillers’ grains had a positive effect on feed conversion ratios and improved immune and gut health status in a broiler [8]. Feeding a total mixed ration containing fermented yellow wine lees reduces oxidative stress and improves milk oxidation in heat-stressed cows [9]. These studies indicate that FDG has long-term prospects for feed substitution in the livestock industry. Microbiome analysis not only allows for a mechanistic assessment of the relationship between animals and their microbial symbionts [10], but also identifies potential markers and metabolic changes in biochemical systems and metabolic changes that can reveal mechanistic links between microbiome composition and abnormal diseases [11]. Often, changes in microbial community composition are accompanied by changes in the metabolic profiles. The interaction between beneficial metabolites and intestinal epithelial cells helps to maintain the intestinal mucosal barrier, thereby reducing the risk of gut-associated infections and disease [12]. The mammalian gut microbial community forms a complex mutualism with its host, which has profound implications for overall health. Therefore, elucidating the biological pathways underlying these characteristic metabolites can help explain the growth promotion mechanism of livestock.

Guanling cattle are one of the 78 local livestock and poultry breeds under key protection in China, one of the “five famous cattle” in China, and the top one of the “four famous cattle” in Guizhou [13]. In November 2016, the former Ministry of Agriculture of the People’s Republic of China officially approved the registration and protection of agricultural geographical indications for “Guanling cattle”(AGI2016-03-1987).

Previous studies have mainly focused on the effects of FDG on the growth performance, feed conversion, and digestibility of livestock and poultry. Our previous studies also showed that adding 30% FDG to the diet did not affect the growth performance, meat quality, and blood metabolism of fattening cattle [14]. However, there is a lack of information on the effects of dietary fermented distiller’s grains supplementation on the gastrointestinal microbiota and metabolites of beef cattle. Therefore, in this study, by combining 16S rDNA sequencing technology and LC-MS detection technology, we analyzed the changes in gastrointestinal microbiota and metabolites in Guanling cattle fed fermented distilling grains diets. These results will provide a reference for the resource utilization of fermented distilling grains and follow-up research on beef cattle production.

## 2. Materials and Methods

### 2.1. Animal Ethics

The animal care and experimental procedures related to this study were authorized by the Experimental Animal Ethics Committee of Guizhou University (Ethics number: EAE-GZU-2020-E018) in accordance with the Chinese guidelines for the ethical treatment of laboratory animals.

### 2.2. Preparation of FDG

The DGs used in this study were from Kweichow Moutai Company Limited (Renhuai, China). The main ingredients of Moutai DG are distilled sorghum and wheat, which are a byproduct of the brewing processes [14]. The microbial starter is from Yijiayi Bioengineering (Shijiazhuang, China), mainly composed of Bacillus, Lactobacillus, Saccharomyces, and Bifidobacterium, as well as various compound enzyme preparations such as amylase, protease, cellulase, and lipase. The fermentation substrate containing 92% (m/m) fresh DGs, 3% (m/m) corn meal, 3% (m/m) rapeseed meal, and 2% (m/m) wheat bran was evenly mixed. Then, the microbial starter was activated with brown sugar water (5 kg of water and 0.5 kg of brown sugar per 100 g of biological fermentation strain) for 3 to 5 days to obtain fermentation broth. Finally, the fermentation broth was sprayed on the surface of the fermentation substrate, mixed well, and sealed for 15 d to generate the FDG [14] used in this study.

### 2.3. Animals, Experimental Design, and Sample Collection

Eighteen Guanling cattle in good health (250 ± 25 kg, aged 18 months, and pre-tested negative for Brucella and Foot-and-Mouth Disease Virus infection) were randomly allocated into three groups with varying FDG levels, namely a basal diet with 0% concentrate replaced by FDG (BD), a basal diet with 15% concentrate replaced by FDG (15% FDG), and a basal diet with 30% concentrate replaced by FDG (30% FDG), with 6 replicates in each group. The basic diet was prepared according to the Chinese industry “Beef Cattle Raising Standard” (NY/T 815-2004), and the ratio of roughage to concentrate was 60:40. According to a previous study of our research group, diet composition and nutritional composition analysis are shown in Appendix A [14]. During the trial, beef cattle were fed at 9:00 and 16:30 daily for 75 d (with a pre-feeding period of 15 days and a formal feeding period of 60 days). Cattle were fed free drinking water, and hygiene and daily management were conducted according to routine methods. By the end of the trial, three randomly selected cattle from each group were transferred to the slaughterhouse and were slaughtered by electric shock; the cattle to be executed were prohibited from being fed 12 h before slaughter. Abomasal (labeled as BD-A, 15% FDG-A, 30% FDG-A, respectively) and cecal content (labeled as BD-C, 15% FDG-C, 30% FDG-C, respectively) were separately collected from each cow into 50 mL sterile tubes. The collected 18 samples were quickly frozen in liquid nitrogen for 12 h and subsequently transferred to −80 °C for storage until microbiomic and metabolomic analyses.

### 2.4. 16S rDNA Sequencing and Data Analysis

Genomic DNA was extracted from the abomasal and cecal content using the DNeasy PowerSoil kit (QIAGEN NO12888, New York, NY, USA) as per the manufacturer’s instructions. The concentration of DNA in each sample was determined using a NanoDrop^®^ ND-2000 instrument (NanoDrop T technologies Inc., Dover, DE, USA). Polymerase chain reaction (PCR) assays were conducted with the Tks Gflex DNA polymerase kit (Takara R060B, Kyoto, Japan) using a pair of primers aimed at the V3-V4 region of the bacterial 16S rDNA gene. The forward primer (F) was 5′-TACGGRAGGCAGCAG-3′ and the reverse primer (R) was 5′-AGGGTATCTAATCCT-3′. The reaction system was 30 μL and consisted of: the PCR reaction components comprised of 15 μL of 2xGflex PCR buffer, 1 μL 5 pmol/μL of forward primer, 1 μL 5 pmol/μL of reverse primer, 1 μL of template DNA, 0.6 μL of Tks Gflex DNA polymerase (1.25 U/μL), and 11.4 μL of H_2_O. The reaction consisted of pre-denaturation at 94 °C for 5 min, followed by 26 cycles of denaturation (94 °C, 30 s), annealing (56 °C, 30 s), and elongation (72 °C, 20 s), before a final extension at 72 °C for 5 min. After amplification via PCR, the products were stored at 4 °C and dispatched to Shanghai OE Biotech. Co. Ltd. (located in Shanghai, China) for paired-end sequencing using an Illumina MiSeq PE300 sequencer (manufactured by Illumina, based in San Diego, CA, USA).

Raw sequencing data were saved in FASTQ format. Any sequences with a base quality below 20 and a length less than 50 bp were removed using Trimmomatic software (version 0.35). Overlapping paired-end reads were merged via Flash software (version 1.2.11). The raw sequences were then analyzed using split_libraries software (version 1.8.0) in QIIME. The further analysis only used high-quality merged sequences, which were characterized by the absence of ambiguous base “N”, single-base repeats less than 8, and a read length greater than 200 bp. UCHIME (version 2.4.2) software was used to detect chimeras in the 16S rDNA sequences. After removal, the remaining sequences were grouped into operational classification units (OTUs) based on a similarity threshold of 97%. The RDP classifier v2.2 software was used for species alignment annotation analysis with a threshold set between 0.7–1. By employing alpha (α) diversity measures, encompassing the Chao1 index, Shannon index, and Simpson indexes, we ascertained the richness and diversity of bacterial and protozoan communities. The computation of beta (β) diversity was executed via QIIME v1.8.0 and then presented utilizing R software v2.15.3. The β-diversity was assessed by calculating Bray-Curtis distances and illustrated using principal coordinate analysis (PCoA). Linear discriminant analysis of effect size (LEfSe) was conducted at the genus level using the Kruskal-Wallis (KW) rank sum test. The KW test was conducted with a threshold of significance set to 0.05 and the log-linear discriminant analysis (LDA) score was set at 2.0 as the cut-off value.

### 2.5. LC-MS-Based Metabolomics Detection and Data Analysis 

Metabolite extraction utilized extracted samples, with equal amounts mixed as quality control (QC) samples before analysis via LC-MS. The analysis process employed a combination system featuring the ACQUITY UPLC I-Class ultra-performance liquid chromatography tandem VION IMS Q-T of a high-resolution mass spectrometer from Waters. Analytes were separated using the ACQUITY UPLC BEH C18 (0100 mm 2.1 mm, 1.7 μm) column, with a constant flow rate of 0.4 mL/min and a 0.1% concentration. Chromatographic conditions: The column was set at a temperature of 45 °C and a sample intake of 1 μL was used. The sample mass spectrometry signal was collected using both positive and negative ion scanning modes, with an electronspray ionization (ESI) for the ion source. Mass spectrum conditions included an electrospray capillary voltage of 2.5 kV, an injection voltage of 40 V, a collision voltage of 4 eV, an ion source temperature at 115 °C, a solvent temperature of 450 m, a carrying gas flow of 900 L/h, a scanning range of 50–1000 amu, a scanning time of 0.2 s, and an interval time of 0.02 s.

The HPLC-MS raw data underwent peak extraction, peak alignment, and normalization using Progenesis QI v2.3 software (Nonlinear Dynamics, Newcastle, UK). The results were subsequently consolidated into a data matrix table for further analysis. Multivariate statistical analysis, utilizing unsupervised principal component analysis (PCA) and orthogonal partial least squares discriminant analysis (OPLS-DA), along with *t*-tests to obtain *p*-values (VIP > 1, *p* < 0.05), was employed to detect dissimilar metabolites among the groups. Origin2020 (OriginLab, Northampton, MA, USA) was utilized to generate a volcano map based on log 2 (FC) and log 10 (*p*-value), which facilitated the swift identification of statistically significant metabolites. The differential metabolic pathway was subsequently constructed using the KEGG database.

### 2.6. Correlation Analysis

Metabolites in the abomasum and cecum, with VIP > 1 and *p* < 0.05, were interactively analyzed with the top 15 dominant microflora in species taxonomy. The Spearman correlation between differential species and differential metabolites was analyzed using “corr” in Python (Version 3.8.8), and the resulting correlations were visualized using a correlation heatmap.

### 2.7. Statistical Analysis

The study assessed differences among the three groups using one-way analysis of variance (ANOVA), with statistical significance set at *p* < 0.05. All statistical analyses were conducted using SPSS 26.0 (Chicago, IL, USA). Spearman’s rank test was used to measure correlations. The overall differences among the groups were compared using the Kruskal-Wallis test.

## 3. Results

### 3.1. 16S rDNA Sequencing Results and Microbial Diversity Characteristics

We obtained the raw sequencing reads of the abomasal and cecal contents of each group by 16S rDNA sequencing. After filtering and removing low-quality sequences, there were more than 70,000 clean reads and more than 78% were high-quality sequences in the effective sequences, indicating that the sample quality can meet the requirements for subsequent analysis. We performed the OTUs classification of the high-quality sequence valid tags obtained by the QC according to 97% similarity and showed the number of shared and specific OTUs in each group in Venn diagrams. As shown in Figure 1A,B, the unique OTU numbers of the BD-A, 15% FDG-A, and 30% FDG-A groups were 2565, 2807, and 3534, respectively, and the number of shared OTUs among the three groups was 4509. The number of unique OTUs for the BD-C, 15% FDG-C, and 30% FDG-C groups were 2782, 1679, and 1879, respectively, and the number of shared OTUs among the three groups was 4820. The sequencing of the abomasal and caecal content in each group indicated Good’s coverage index values nearing 1 (Figure 1C,D), implying that the sequencing depth was adequate for examining microbial structures. To investigate the range of microbial communities, we analyzed the α diversity indices at the species level. The results showed that there was no significant difference in Chao1, Shannon, and Simpson indices between 15% FDG-A and 30% FDG-A compared with BD-A (*p* > 0.05) (Figure 1E–G). Compared with BD-C, there were no significant differences in Chao1, Shannon, and Simpson indices between the 15% FDG-C and 30% FDG-C groups (*p* > 0.05) (Figure 1H,J). It showed that feeding FDG had little effect on abomasum and cecum microbiota diversity and abundance in Guanling cattle. To investigate whether the microbiota varied between the different groups, we calculated the Bray-Curtis distance at the species level in each group. As shown in Figure 1K,L, the PCoA revealed significant differences in the abomasal and cecal contents between the groups. The non-metric multidimensional scaling (NMDS) analysis demonstrated significant variation in microbial composition amongst the groups studied, as indicated in Figure 1M,N.

### 3.2. Analysis of the Structural Composition of Microbial Communities in Abomasal and Cecal Contents

To further explore the effects of FDG on the composition of the abomasum and cecum microbiota, we identified the relative abundance of the microbiota at the phylum and genus levels to demonstrate FDG-induced fundamental changes. The three dominant and most abundant microbial phyla in the abomasum were Firmicutes, Bacteroidetes, and Proteobacteria, which together accounted for 88.26%, 90.07%, and 95.29% of the BD-A, 15% FDG-A, and 30% FDG-A groups, respectively. Compared with BD-A, at the phylum level, the relative abundance of Spirochaetes was significantly decreased in the 15% FDG-A group; Proteobacteria and Spirochaetes were significantly decreased while Bacteroidetes were significantly increased in the 30% FDG-A group (Figure 2A). At the genus level, the relative abundance of Treponema_2 in the 15% FDG-A group was significantly decreased, while Ruminococcaceae_UCG-010 was significantly increased. The relative abundance of Treponema_2 was significantly decreased in the 30% FDG-A group, while the abundance of Prevotellaceae_UCG-001 was significantly increased (Figure 2B).

The three dominant and most abundant microbial phyla in the cecum were also Firmicutes, Bacteroidetes, and Proteobacteria, which together accounted for 95.2%, 98.88%, and 98.99% of the BD-C, 15% FDG-C, and 30% FDG-C groups, respectively. Compared with BD-C, at the phylum level, the relative abundance of Proteobacteria and Spirochaetes was significantly lower in the 15% FDG-C group, and the relative abundance of all Proteobacteria, Spirochaetes, and Tenericutes was significantly lower in the 30% FDG-C group (Figure 2C). At the genus level, the relative abundance of Ruminococcaceae_UCG-005 and Bacteroides in the 15% FDG-C group was significantly increased. The relative abundance of Ruminococcaceae_UCG-005 in the 30% FDG-C group was significantly increased, while that of Ruminococcaceae_UCG-010 was significantly decreased (Figure 2D).

Linear discriminant analysis (LDA) and LDA Effect Size (LefSe) were employed to examine the impact of the microbial fermentation of distillers’ grains on the abomasal and caecal microbiota of cattle. As shown in Figure 2E, the most influential bacterial community structure in the BD-A group was HIMB11, Gammaproteobacteria. In the 30% FDG-A group, Bacteroidetes, Prevotellace_UCG_003, and Ruminococcaceae_UCG_007 had the greatest impact (all LDA scores (log10) > 3). In the 15% FDG-A group, Riemeralla, Ruminococcaceae_UCG_002, and Sediminispirochaeta had the greatest impact (all LDA scores (log10) > 3). As shown in Figure 2F, the bacterium with the greatest impact in the 30% FDG-C group was Prevotella_1(LDA score (log10) > 3). These data suggest that these differentially abundant microbiota are sufficient to distinguish the microbiota of the abomasum and cecum in the three groups.

### 3.3. Metabolomics Analysis of Abomasal and Caecal Contents

Feeding probiotics FDG shaped the gastrointestinal microbiota of Guanling cattle, and in turn, these microbes had a positive effect on the gastrointestinal content metabolism.

To further investigate the metabolic alterations of microbiota in Guanling cattle when supplemented with various doses of FDG, abomasal and caecal samples were analyzed using LC-MS. The metabolic profiles of the different groups were compared using supervised OPLS-DA to assess overall differences. OPLS-DA plots showed that BD-A vs. 15% FDG-A, BD-A vs. 30% FDG-A, 15% FDG-A vs. 30% FDG-A, BD-C vs. 15% FDG-C, BD-C vs. 30% FDG-C, and 15% FDG-C vs. 30% FDG-C had obvious separation effects (Figure 3A–F).

In this study, compounds with a VIP value > 1 in the first principal component of the OPLS-DA model and a *p*-value < 0.05 in the *t*-test were identified as differential metabolites. The larger the VIP value, the greater the contribution of variables to grouping. The screening results of differential metabolites in each group are shown in the volcano diagram. Compared to the BD-A group, 36 (all down-regulated) and 52 (17 up-regulated and 35 down-regulated) differential metabolites were screened in the 15% FDG-A (Figure 4A) and 30% FDG-A groups (Figure 4B), respectively. Compared with the 15% FDG-A group, 17 biologically significant differential metabolites were screened out in the 30% FDG-A group (Figure 4C). Compared to BD-C, twenty-six (eighteen up-regulated and eight down-regulated) and twenty-nine (ten up-regulated and nineteen down-regulated) differential metabolites were screened in the 15% FDG-C group (Figure 4D) and 30% FDG-C group (Figure 4E), respectively. Twenty-nine biologically significant differential metabolites were screened out in the 30% FDG-C group compared with the 15% FDG-C group (Figure 4F). These differential metabolites mainly included lipids and lipid-like molecules, organoheterocyclic compounds, organic acids and derivatives, alkaloids and derivatives, etc.

Performing pathway enrichment analysis on differential metabolites assists in comprehending the mechanism of metabolic pathway changes in distinct samples. To obtain metabolic pathway enrichment results, a metabolic pathway enrichment analysis of differential metabolites was conducted using the KEGG database. Compared with the BD-A group, the metabolic pathways significantly enriched in the 15% FDG-A group were primary bile acid biosynthesis, bile secretion, cholesterol metabolism, insulin resistance, and taurine and hypotaurine metabolism (Figure 5A); the metabolic pathways significantly enriched in the 30% FDG-A group were primary bile acid biosynthesis, bile secretion, cholesterol metabolism, taurine and hypotaurine metabolism, and linoleic acid metabolism (Figure 5B). KEGG pathway enrichment analysis showed that 15% FDG-A and 30% FDG-A had four shared metabolic pathways compared with BD-A. Compared with the 15% FDG-A group, 11 metabolic pathways were significantly enriched in the 30% FDG-A group (*p* < 0.05), mainly including: purine metabolism, melanogenesis, cocaine addiction, amphetamine addiction, alcoholism, pantothenate and CoA biosynthesis, etc. (Figure 5C).

Compared with the BD-C group, the metabolic pathways significantly enriched in the 15% FDG-C group were choline metabolism in cancer, fatty acid elongation, primary bile acid biosynthesis, fatty acid degradation, and glyceropho6spholipid metabolism (Figure 5D). Additionally, the differential metabolites in the 30% FDG-C group were not enriched to metabolic pathways. Compared with the 15% FDG-C group, 19 metabolic pathways were significantly enriched in the 30% FDG-C group (*p* < 0.05), mainly including: tyrosine metabolism, melanogenesis, cocaine addiction, amphetamine addiction, the prolactin signaling pathway, the HIF-1 signaling pathway, etc. (Figure 5E).

The differences in metabolites that could be significantly enriched to the metabolic pathways in the groups are shown in Table 1.

### 3.4. Correlation Analysis between Microbiome and Metabolome in Abomasal Content and Caecal Content

To determine the potential interactions between the abomasum and cecum microbiota and their metabolites, Spearman’s correlation calculation method was used to draw the correlation heat map of the association analysis results between the differential metabolites enriched in metabolic pathways and the TOP 15 microbiota. As shown in the figure, compared with the BD-A group, Ruminococcaceae_UCG-010 was negatively correlated with cholic acid, taurocholic acid, chenodeoxycholic acid, and adenine in the 15% FDG-A group (Figure 6A). Treponema-2, the relative abundance of which was significantly reduced in the 30% FDG-A group, showed a significant negative correlation with 9, 12, 13-trihome, Hypoxanthine, and a significant positive correlation with Taurocholic acid (Figure 6B). Compared with the 15% FDG-A group, hypoxanthine in the 30% FDG-A group was significantly positively correlated with Prevotellaceae_UCG-001, Prevotellaceae_UCG-003, and Prevotella_1, and significantly negatively correlated with Treponema_2. Cholic acid was negatively correlated with [Eubacterium]_coprostanoligenes_group, Ruminococcaceae_UCG-010, and Prevotellaceae_UCG-001 (Figure 6C). There was a significant positive correlation with Fibrobacter. Compared with the BD-C group, Bacteroides were significantly positively correlated with Palmitic acid and LysoPC (18:1(11Z)) in the 15% FDG-C group (Figure 6D). Compared with the 15% FDG-C group, the Lachnospiraceae_NK4A136_group was significantly positively correlated with L-Lactic acid and Rosmarinic acid, and significantly negatively correlated with L-Tyrosine in the 30% FDG-C group (Figure 6E).

## 4. Discussion

The DGs are a kind of high-quality feed resource, and their nutritional value can be improved by microbial fermentation [5]. The FDG contain a variety of beneficial microorganisms, which can improve intestinal microecology due to their nutrient decomposition and modification of intestinal microbiome, thus promoting digestion and absorption in the body [15,16], thereby improving production efficiency and ensuring animal health [17].

The high-throughput sequencing of 16S rDNA amplicons was used to understand the changes in the microbiota of the abomasum and cecum of Guanling cattle after the replacement of concentrate with different FDG levels. The results contribute to our understanding of the potential beneficial effects of FDG on cattle. In the present study, we found no significant effect of FDG feeding on alpha diversity in the bovine abomasum and cecum, indicating that microbial diversity responds in a relatively stable way to FDG, and similar observations were also reported by Huan Li et al. [18]. The composition of the microbiota plays an important role in the function of the GI tract. In this study, we found that the top three dominant microorganisms in the abomasum and cecum of Guanling cattle after feeding FDG were Firmicutes, Bacteroidetes and Proteobacteria at the phylum level, consistent with the findings of Mariano A et al. in the rumen microbiota of dairy cows [19]. By comparing the microbial community structure of rumen and abomasum, it was found that after feeding fermented lees, the microbial community structure of rumen and abomasum significantly changed. The main points of interaction were the significant increase in Bacteroidetes and Prevotella, and the significant decrease in Spirochetes and Tenericutes in the rumen and abomasum. The difference was that Fibrobacter, Ruminococcaceae_UCG-010, and Erysipelotrichosiaceae _UCG-004 in the rumen were significantly reduced. While Proteobacteria in the abomasum was significantly decreased, Ruminococcaceae_UCG-010 significantly increased. Bacteroidetes and Firmicutes contain a large number of fiber-degrading bacteria that degrade cellulose in distillers’ grains feed, Firmicutes provide energy through the fermentation of polysaccharides into short-chain fatty acids [20], and also promote energy absorption and fat deposition [21]; however, Bacteroidetes mainly play an important role in the degradation of carbohydrates and are known for degrading starch, pectin, and xylan. In the present study, Bacteroidetes were significantly higher and Firmicutes tended to be higher in the 30% FDG-A group compared to the BD-A group, indicating that dietary FDG could promote host uptake or the storage of energy, and thus promote ruminant growth and development. At the genus level, Ruminococcaceae was the most dominant bacterial genus in bovine cecum. Compared with BD-C, the abundance of Ruminococcaceae_UCG-005 in the 15% FDG-C and 30% FDG-C groups was significantly increased. This is in general agreement with previous studies, in which ruminococcaceae were the most abundant bacteria in cow fecal bacteria regardless of diet treatment [22,23]. The genus Ruminococcus is now considered to be an important contributor to the intestinal ecosystem, producing large amounts of cellulases and hemicellulases that can degrade complex polysaccharides and convert them into various host nutrients [24]. The increased abundance of Ruminococcus may also be related to the regulation of blood glucose levels. In addition to Ruminococcaceae_UCG-005, the abundance of Prevotellaceae_UCG-003 and Bacteroides in the gastrointestinal tract was significantly increased after adding FDG to the diet. Prevotellaceae_UCG-003 belongs to the genus Prevotella, which has the common characteristic of regulating intestinal inflammation potential [25]. Meanwhile, Prevotella species produce a variety of enzymes, such as glycoside hydrolases and polysaccharide lyases, capable of degrading dietary fiber in plant cell walls. Bacteroides is also a genus that degrades polysaccharides and has previously been shown increase in abundance in obese animals fed a high-fiber diet [26]. Moreover, members of the genus Bacteroides have been shown to synthesize conjugated linoleic acid, known to have anti-diabetic, anti-atherosclerotic, anti-obesity, lipid-lowering, and immunomodulatory properties [27,28,29]. These results indicate that the bacteria significantly increased in the gastrointestinal tract belong to the bacteria with strong fiber degradation ability, which may be due to the high fiber content of distiller’s grains in the diet. Fiber can induce an increase in fiber-degrading bacteria, promote VFA metabolism, and help improve the feed utilization efficiency of the host, thereby promoting the growth and development of Guanling cattle.

Some potential prebiotics can stimulate the growth of beneficial bacteria and develop resistance to pathogenic bacteria. In this study, in addition to the significant increase in the abundance of the above beneficial bacteria, we also found that the relative abundance of potentially pathogenic phylum Proteobacteria and Spirochaetes in the abomasum and cecum of cattle was significantly decreased after feeding FDG. Proteobacteria are associated with intestinal inflammation, increased risk of disease, and have been identified by some authors as possible markers of microbiome instability. Arula-7 powder was shown to reduce the incidence of diarrhea caused by E. coli by reducing the abundance of Proteobacteria [30], which is similar to the study in this paper. Spirocholytes are spiral-like, some of which are pathogenic to humans and animals [31] and are well-known diarrheal pathogens in veterinary medicine. These results suggest that 15% FDG and 30% FDG have a beneficial effect on the balance of gastrointestinal microbiota in cattle, and that FDG can reduce the risk of infectious diseases by reducing the abundance of Proteobacteria and Spirochaetes.

Bacteria in the gastrointestinal tract of Guanling cattle produce various metabolites that mediate host metabolism. We used untargeted metabolomics to analyze the effects of feeding probiotics to FDG on gastrointestinal metabolites in Guanling cattle. In our study, we found that metabolites such as chenodeoxycholic acid, cholic acid, and Taurocholic acid were significantly down-regulated in the 15% FDG and 30% FDG groups, all of which are involved in the primary bile acid biosynthesis pathway, and chenodeoxycholic acid and cholic acid are the end products of primary bile acids. Bile acids are generally considered carcinogenic substances, and studies have shown that Taurocholic acid is the main bile acid component in the reflux fluid of gastroesophageal reflux disease [32], which can promote the progression of esophageal adenocarcinoma [33]. Chenodeoxycholic acid can induce oxidative stress, DNA damage, and inflammation, leading to the carcinogenesis of esophageal adenocarcinoma [34]. Other studies have found that bile acids are increased in non-alcoholic fatty liver disease [35], and the serum chenodeoxycholic acid level is higher in breast cancer patients. These results suggest that feeding FDG may have a beneficial effect on cholestasis, and can inhibit the occurrence of cancer and tumors in Guanling cattle to a certain extent. The effect of bile acids on gut microbes is complex, and there is a strong association between changes in bile acids and gut bacteria. The results of the Spearman analysis showed that chenodeoxycholic acid, cholic acid, and taurocholic acid were negatively correlated with Prevotellaceae_UCG-001 and Ruminococcaceae_UCG-010. These results indicated that the decrease in these metabolites in Guanling cattle after feeding FDG might be due to the increase in Prevotellaceae_UCG-001 and Ruminococcaceae_UCG-010 abundance. Studies have shown that the excessive accumulation of adenine will be converted to 8-dihydroxyadenine and, eventually, to 2, 8-dihydroxyadenine. The crystallization of these insoluble substances in the renal tubules will cause damage and can induce serious renal harm [36]. In the present study, adenine metabolites were decreased in the 15% FGD-A group, indicating that feeding FDG did not impair renal function in cattle and may alleviate adenine-induced chronic kidney disease by regulating the purine metabolism signaling pathway.

We also found that metabolites such as palmitic acid and 9, 12, 13-trihome were significantly increased in the experimental group supplemented with FDG. Palmitic acid, an ACOX1 (metabolic enzyme acyl-coa) oxidase substrate and major fatty acid in high-fat diets, has been shown to produce energy and regulate intracellular signaling molecules involved in cancer development. It exerts anticancer effects by activating endoplasmic reticulum (ER) stress/ER calcium release/transferrin-dependent ferroptosis [37] and promotes the metastasis of melanoma, breast cancer, and gastric cancer in a CD36-dependent manner [38]. Spearman’s correlation analysis showed that palmitic acid was negatively correlated with Proteobacteria and positively correlated with Prevotella-1. Furthermore, 9, 12, 13-TRIhome is the end product of linoleic acid metabolism, which has been shown to have various effects on the body, including reducing inflammation, regulating metabolism, and delaying aging, in addition to improving metabolic fitness, preventing fatigue, and stimulating memory-like phenotypes with superior effector functions [39]. Many metabolites of linoleic acid are potent lipid mediators. TriHOME is made up of linoleic acid-derived oxylipins with potential physiological correlates in inflammatory processes as well as in maintaining an intact skin barrier [40]. Moreover, 9, 12, 13-Trihome has been described as a potentially useful adjuvant in influenza vaccine preparations [41]. The results suggest that feeding FDG can reduce gastrointestinal inflammation and inhibit cancer development in Guanling cattle, which further indicates that feeding FDG has beneficial effects on the bodies of the cattle.

Finally, we found that L-Tyrosine, a differential metabolite, was significantly upregulated in both the abomasum and cecum in the 30% FDG group compared with the 15% FDG group. It was significantly enriched in multiple negative metabolic pathways, such as melanogenesis, cocaine addiction, amphetamine addiction, alcoholism, Parkinson’s disease, etc. L-Tyrosine is the starting material for melanin biosynthesis [42], and the production of melanin may produce more reactive oxygen species, which may accelerate the occurrence of cancer [43]. Cocaine is a highly addictive drug, and the physical and mental consequences of cocaine addiction are serious and clinically relevant [44]. Amphetamine use is associated with psychosis [45]. Alcoholism is a well-known risk factor for the development of a variety of cancers, including esophageal cancer and colon cancer, and is also associated with potentially fatal diseases such as cirrhosis, dyslipidemia, or malnutrition [46]. Parkinson’s disease is also a neurodegenerative disease [47]. There are indications that at 30% FDG feeding, potentially inflammatory compounds released into the abomasum and cecum lead to changes in metabolic pathways that may cause harm to the overall health of Guanling cattle. However, we found that Rosmarinic acid, the differential metabolite in 30% FDG-C compared with 15% FDG-C, is a natural polyphenol compound with various biological activities, such as antioxidant, anti-inflammatory, antibacterial, and anticancer activities [48]. The differential metabolites 3-(3-indolyl)-2-oxopropanoic acid and Pantothenic acid were up-regulated in the 30% FDG-A group compared with the 15% FDG-A group. Moreover, they were enriched in tryptophan metabolism, Pantothenate, and CoA biosynthesis, respectively. Studies have shown that tryptophan metabolism and its indole metabolites play an important role in depression and the alleviation of different types of diseases, such as neurological diseases and cancer [49,50]. Meanwhile, Pantothenic acid is an essential anti-pellagra vitamin for the growth and health of many animals [51]. These results suggested that the body was adjusting itself and trying to resist the influence of some adverse factors, so that no obvious disease occurred in Guanling cattle during our feeding process. In general, from the metabolomic point of view, in production practice, to avoid unnecessary losses, it is recommended to add about 15% of fermented distilled grains when feeding beef cattle.

## 5. Conclusions

The results of this study indicated that FDG replacement feeding to some extent improves the GI microbial community structure in Guanling cattle. FDG feeding increased the abundance of bacteria that promote glycolysis and degradable fiber in the GI tract of Guanling cattle, while reducing the abundance of some opportunistic pathogenic bacteria, and these changes had positive effects on the metabolism (e.g., linoleic acid metabolism, primary bile acid biosynthesis) of GI contents, thereby promoting the GI health of Guanling cattle. Comprehensive analysis of non-target metabolism showed that it was feasible to replace 15% concentrate with FDG, and these findings provided a theoretical basis for the large-scale application of FDG as a feed resource and subsequent research on beef cattle production.

## Figures and Tables

**Figure 1 animals-13-03437-f001:**
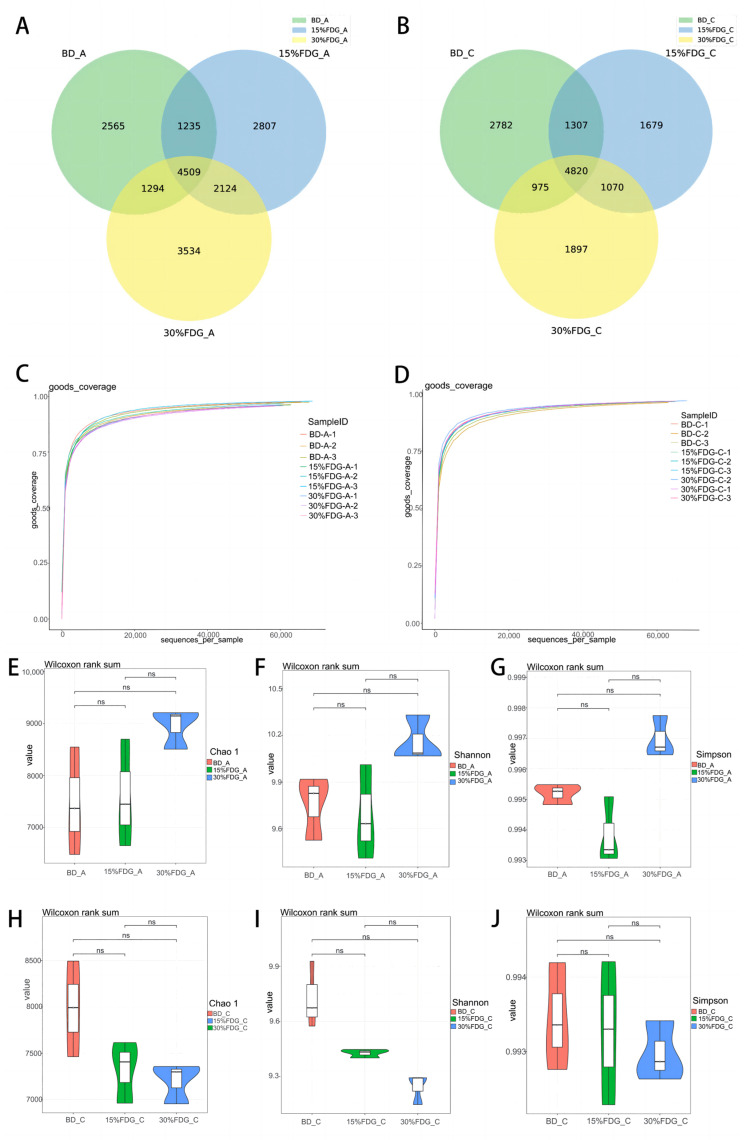
Diversity and composition of microbes in different groups. OTU comparative analysis of gastrointestinal microbiota of Guanling cattle (**A**) abomasum and (**B**) cecum. Good’s coverage index dilution curve from microbiota in the (**C**) abomasum and (**D**) cecum of Guanling cattle. Alpha diversity analysis of abomasum using the (**E**) Chao1, (**F**) Shannon, and (**G**) Simpson indices. Alpha diversity analysis of cecum using the (**H**) Chao1, (**I**) Shannon, and (**J**) Simpson indices. Labeled ns indicates no significant difference. (**K**) Abomasal and (**L**) cecal principal coordinate analysis was carried out using the Bray-Curtis distance matrix. Horizontal non-metric multidimensional scale (NMDS) in the (**M**) abomasum and (**N**) cecum. BD: basal diet group; 15% FDG: a basal diet with 15% concentrate replaced by FDG; 30% FDG: a basal diet with 30% concentrate replaced by FDG. Note: marked “ns” indicates no difference.

**Figure 2 animals-13-03437-f002:**
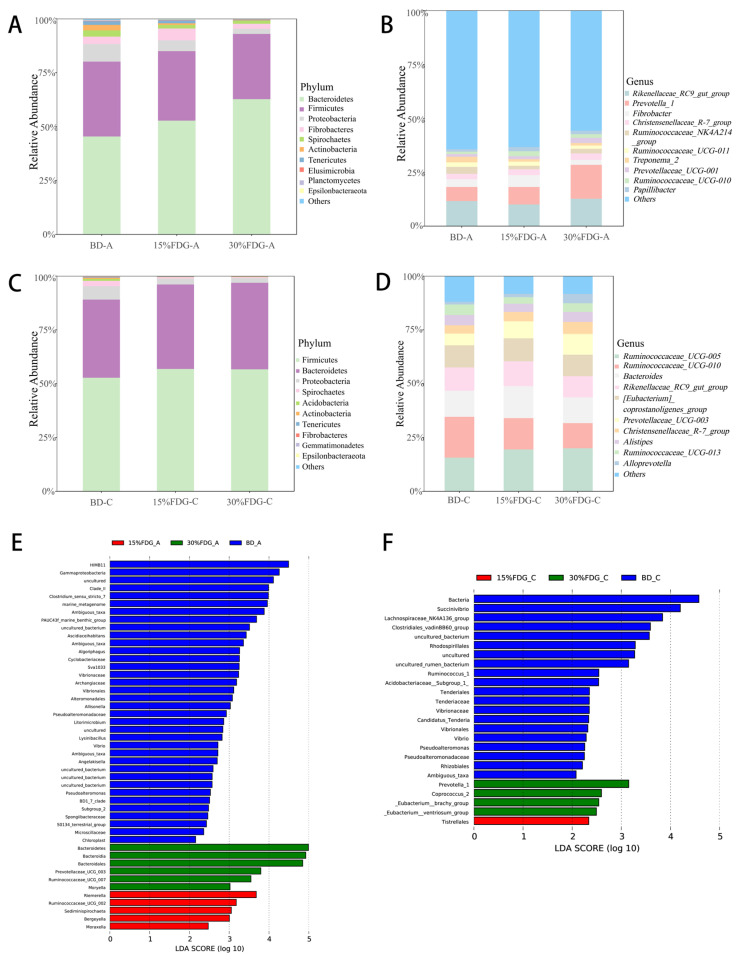
Composition analysis of Guanling cattle abomasum and cecum microorganisms and differential microbiological analysis among different groups. Microbial composition profiles of three abomasal treatment groups at phylum (**A**) and genus (**B**) levels. Microbial composition profiles of three cecal treatment groups at phylum (**C**) and genus (**D**) levels. LDA Effect Size (LEfSe) analysis of abomasum groups (**E**). LEfSe analysis of cecum groups (**F**). BD: basal diet group; 15% FDG: a basal diet with 15% concentrate replaced by FDG; 30% FDG: a basal diet with 30% concentrate replaced by FDG.

**Figure 3 animals-13-03437-f003:**
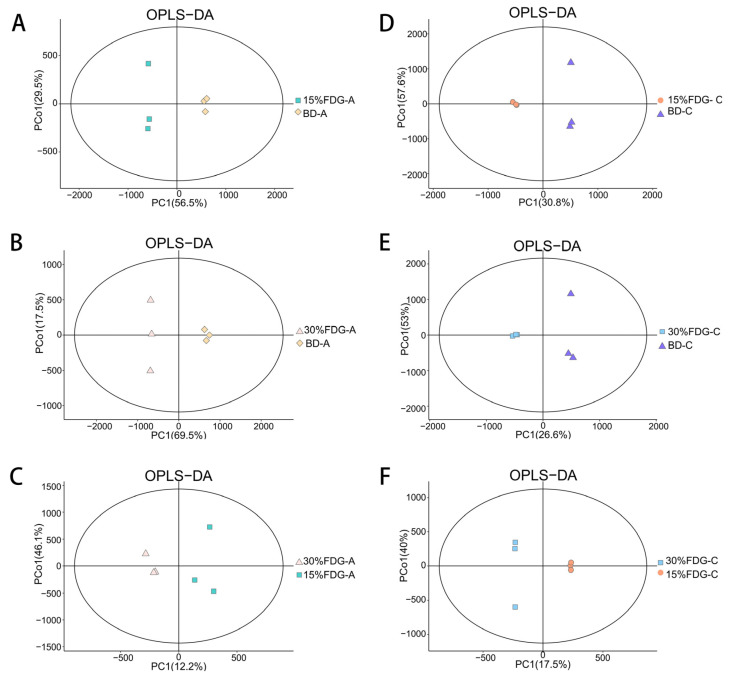
OPLS-DA analysis of abomasum metabolites for (**A**) 15% FDG-A vs. BD-A, (**B**) 30% FDG-A vs. BD-A, and (**C**) 30% FDG-A vs. 15% FDG-A, respectively. OPLS-DA analysis of cecum metabolites for (**D**) 15% FDG-C vs. BD-C, (**E**) 30% FDG-C vs. BD-C, and (**F**) 30% FDG-C vs. 15% FDG-C, respectively. BD: basal diet group; 15% FDG: a basal diet with 15% concentrate replaced by FDG; 30% FDG: a basal diet with 30% concentrate replaced by FDG.

**Figure 4 animals-13-03437-f004:**
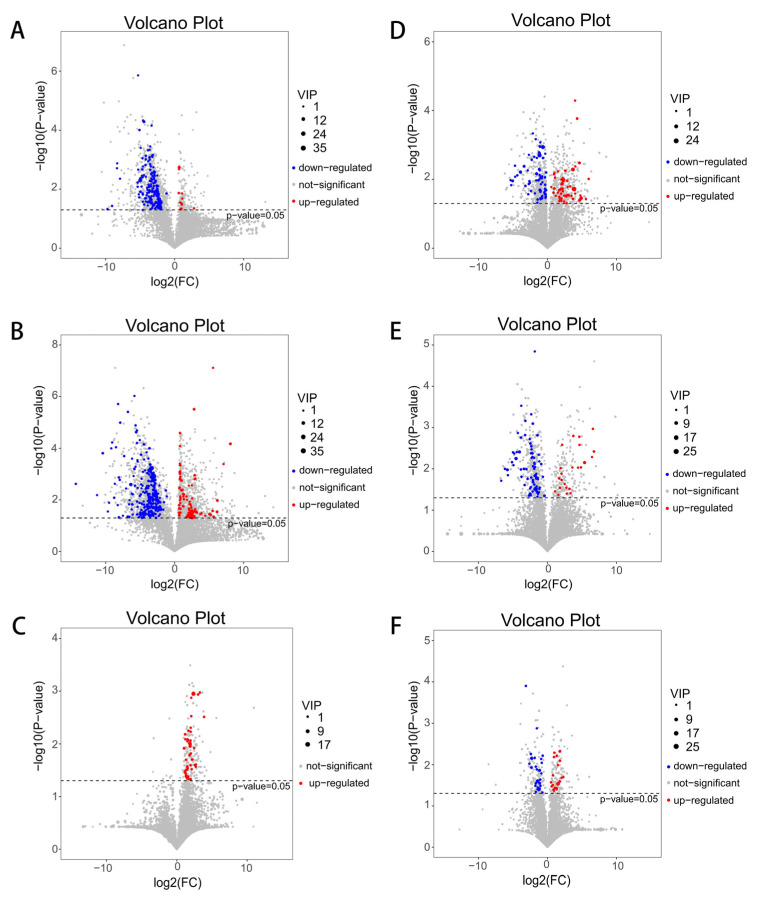
Volcano plots. The volcano plot was generated based on abomasum metabolites detected by the untargeted analysis in (**A**) 15% FDG-A vs. BD-A, (**B**) 30% FDG-A vs. BD-A, and (**C**) 30% FDG-A vs. 15% FDG-A. The volcano plot was generated based on cecum metabolites detected by the untargeted analysis in (**D**) 15% FDG-C vs. BD-C, (**E**) 30% FDG-C vs. BD-C, and (**F**) 30% FDG-C vs. 15% FDG-C. In the volcano plots, the red origin represents significantly up-regulated metabolites in the experimental group, the blue origin represents significantly down-regulated metabolites, and the gray point represents insignificant metabolites. The dotted line indicates the significance level threshold (*p* = 0.05).

**Figure 5 animals-13-03437-f005:**
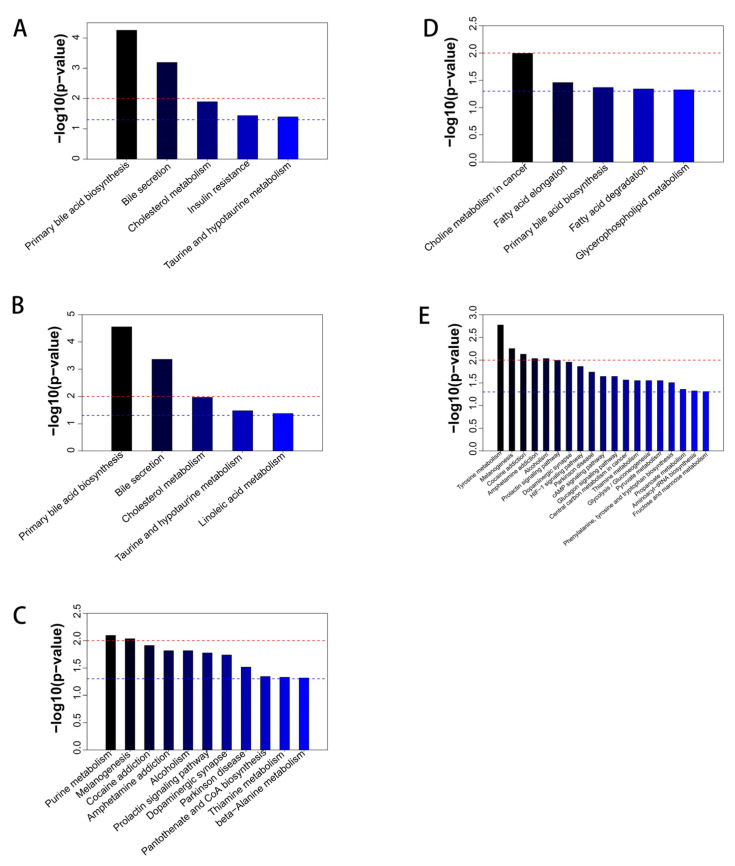
Enrichment map of metabolic pathways. The enrichment map of metabolic pathways is used to show the different abomasum metabolic pathways in (**A**) 15% FDG-A vs. BD-A, (**B**) 30% FDG-A vs. BD-A, and (**C**) 30% FDG-A vs. 15% FDG-A. The enrichment map of metabolic pathways was used to show the different cecum metabolic pathways in (**D**) 15% FDG-C vs. BD-C and (**E**) 30% FDG-C vs. 15% FDG-C. In the enrichment map of metabolic pathways, the red dotted line represents a *p*-value = 0.01, and the blue dotted line represents a *p*-value = 0.05. When the top of the bar is beyond the blue line, it indicates that the signaling pathway it represents is significant.

**Figure 6 animals-13-03437-f006:**
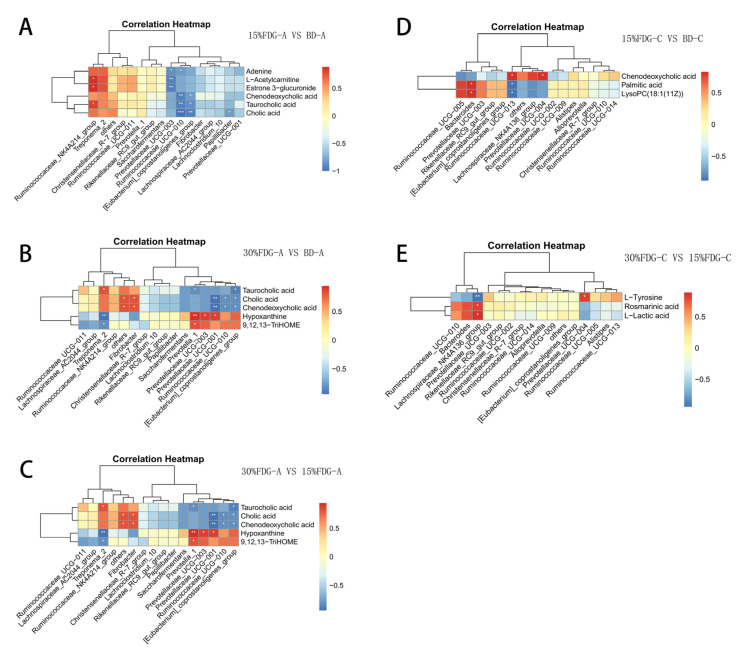
Correlation between metabolites enriched into metabolic pathways and TOP15 microbiota. Spearman’s level correlation between cecum microflora and metabolites. Correlation analysis between abomasum dominant microbials and differential metabolites in (**A**) 15% FDG-A vs. BD-A, (**B**) 30% FDG-A vs. BD-A and (**C**) 30% FDG-A vs. 15% FDG-A. Correlation analysis between cecum dominant microbials and differential metabolites in (**D**) 15% FDG-C vs. BD-C and (**E**) 30% FDG-C vs. 15% FDG-A. In the figure, red and blue colors represent positive and negative correlations, respectively, and color gradation indicates the size of the correlation coefficient. * *p* < 0.05, ** *p* < 0.01, and *** *p* < 0.001 indicate significant differences between the microbiota and metabolites.

**Table 1 animals-13-03437-t001:** Differential metabolites enriched metabolic pathways in the 15% FDG group and the 30% FDG group.

Nos.	Metabolites	VIP	*p*-Value	FC	Annotation
15%FDG-A vs. BD-A					
1	Adenine	1.023	<0.001	0.025	Purine metabolism
2	Chenodeoxycholic acid	2.119	0.037	0.002	Primary bile acid biosynthesis
3	Cholic acid	13.380	0.005	0.031	Primary bile acid biosynthesis|Bile secretion
4	Estrone 3-glucuronide	1.037	0.026	0.072	Steroid hormone biosynthesis
5	L-Acetylcarnitine	1.637	0.008	0.094	Insulin resistance
6	Taurocholic acid	41.737	0.028	0.230	Primary bile acid biosynthesis| Bile secretion|Cholesterol metabolism| Taurine and hypotaurine metabolism
30%FDG-A vs. BD-A					
1	9, 12, 13-TriHOME	1.104	0.003	4.171	Linoleic acid metabolism
2	Chenodeoxycholic acid	1.779	0.041	0.033	Primary bile acid biosynthesis
3	Cholic acid	9.674	0.023	0.252	Primary bile acid biosynthesis|Bile secretion
4	Hypoxanthine	3.987	<0.001	7.063	Purine metabolism
5	Taurocholic acid	40.776	0.001	0.119	Primary bile acid biosynthesis| Bile secretion|Cholestero lmetabolism| Taurine and hypotaurine metabolism
30%FDG-A vs. 15%FDG-A					
1	3-(3-Indolyl)-2-oxopropanoic acid	1.220	0.046	2.822	Tryptophan metabolism
2	Guanine	1.417	0.006	3.135	Purine metabolism
3	Hypoxanthine	10.526	0.001	5.143	Purine metabolism
4	L-Tyrosine	1.740	0.029	2.901	Melanogenesis|Cocaine addiction|Amphetamine addiction|Alcoholism|Prolactin signaling pathway|Dopaminergic synapse|Parkinson disease|Thiamine metabolism|Phenylalanine, tyrosine and tryptophan biosynthesis|Aminoacyl-tRNA biosynthesis|Phenylalanine metabolism|Ubiquinone and other terpenoid-quinone biosynthesis|Tyrosine metabolism
5	Pantothenic acid	1.385	0.027	6.518	Pantothenate and CoA biosynthesis|beta-Alanine metabolism|Vitamin digestion and absorption
15% FDG-C vs. BD-C					
1	Chenodeoxycholic acid	13.983	0.002	0.448	Primary bile acid biosynthesis
2	LysoPC(18:1(11Z))	1.126	0.011	4.981	Choline metabolism in cancer| Glycerophospholipid metabolism
3	Palmitic acid	3.628	0.021	3.297	Fatty acid elongation|Fatty acid degradation| Fatty acid biosynthesis| Biosynthesis of unsaturated fatty acids
30% FDG-C vs. 15% FDG-C					
1	L-Lactic acid	1.057	0.046	0.310	HIF-1 signaling pathway|cAMP signaling pathway|Glucagon signaling pathway|Central carbon metabolism in cancer|Glycolysis/Gluconeogenesis|Pyruvate metabolism|Propanoate metabolism|Fructose and mannose metabolism
2	L-Tyrosine	1.343	0.031	1.697	Tyrosine metabolism|Melanogenesis|Cocaine addiction|Amphetamine addiction|Alcoholism|Prolactin signaling pathway|Dopaminergic synapse|Parkinson disease|Thiamine metabolism|Phenylalanine, tyrosine and tryptophan biosynthesis|Aminoacyl-tRNA biosynthesis|Phenylalanine metabolism|Ubiquinone and other terpenoid-quinone biosynthesis
3	Rosmarinic acid	1.981	0.028	0.309	Tyrosine metabolism

Note: VIP, variable influence on projection; FC, fold change; Nos., numbers.

## Data Availability

The authors confirm that the data supporting the study findings are available in the article and Appendix A.

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
