# Peer review of "Partially Alternative Feeding with Fermented Distillers’ Grains Modulates Gastrointestinal Flora and Metabolic Profile in Guanling Cattle"

_animals, 2023, doi:10.3390/ani13223437_

Round 1
Reviewer 1 Report
Comments and Suggestions for Authors
Comments on abstract, title, and introduction.
- The title is relevant to the aim of the study. This study has a good idea to study conducted to investigate the effects of partial concentrate replacement by FDG feeding in Guanling cattle.
## Comments on the introduction: the topic of this paper is clear. Moreover, the research question sightly appears.
## Comments on methodology
- the methods used in this study were valid and reliable
- I am satisfied with the use of a good number of 18 to collect samples.
But I have some comments on this point:
- I suggested that: the authors support the manuscript with a table of the chemical composition of FDG and all group diets after adding FGD.
- I have a question why the authors don’t collect samples from the rumen to give more explanations? Like correlation between the end product of rumen fermentation and 16S rDNA sequencing of microbes (example: Methanogens) and rumen metabolites.
## Comments on data results and discussion
The paper is generally well-written and structured.
The main results of this study:
My comment here:
Figure 1 Diversity and composition of microbes in different groups. Need more resolution or change the design.
## Discussion:
You need to go deeper with the scientific explanation of your findings and add more citations that support your results especially. I suggest make diagram to explain the correlation between feeding with fermented distillers' grains modulates gastrointestinal flora and metabolic pathways.
## Comments on conclusions
- the conclusions answer the aims of the study. To investigate the effects of partial concentrate replacement by FDG feeding in Guanling cattle.
- But try to focus on your findings and shorten this section and recommend the best percentage of FDG which can partially be alternative in the ruminants feeding
- I think scientists have opportunities to inform future research and more research is needed.
Author Response
Response to Reviewer 1 Comments
Dear Reviewer 1,
Thank you very much for your valuable comments, and our point-to-point responses are as follows:
Point 1: I suggested that: the authors support the manuscript with a table of the chemical composition of FDG and all group diets after adding FDG.
Response 1: Thank you very much for your comments. We have presented these data in the Supplementary material and have also cited a reference, which is previously published by our research team[1]. Since these data have already been published by other members of our group, it is not convenient to add them to our manuscript. However, we can present it here for review.
|
Supplementary Table |
||||
|
TABLE S1 | Composition and nutritional content of experimental diets varying in levels of fermented distillers’ grain (FDG) |
||||
|
Items |
BD |
15%FDG |
30%FDG |
FDG |
|
Ingredient, % DM inclusion |
||||
|
Pennisetum Sinese Roxb |
60 |
60 |
60 |
|
|
Groundcorn |
20 |
10 |
0 |
|
|
Soybean meal |
6 |
3 |
0 |
|
|
Rapeseed meal |
4 |
2 |
0 |
|
|
Wheat bran |
8 |
8 |
8 |
|
|
FDG |
0 |
15 |
30 |
|
|
Supplement |
|
|
|
|
|
Rock fine |
1.4 |
1.4 |
1.4 |
|
|
Salt |
0.05 |
0.05 |
0.05 |
|
|
Calcium bisulfate |
0.42 |
0.42 |
0.42 |
|
|
Sodium sulfate |
0.08 |
0.08 |
0.08 |
|
|
Microelement additive |
0.05 |
0.05 |
0.05 |
|
|
Total |
100 |
100 |
100 |
|
|
Chemical analysis |
|
|||
|
Dry matter (%) |
70.58 |
70.68 |
71.2 |
31.5 |
|
Gross energy (MJ/kg DM) |
15.76 |
16.6 |
16.7 |
18.69 |
|
Crude protein (% DM) |
11.4 |
12.64 |
13.88 |
21.88 |
|
Neutral detergent fiber (% DM) |
54.47 |
55.47 |
56.47 |
36.56 |
|
Acid detergent fiber (% DM) |
29.37 |
32.96 |
36.54 |
23.33 |
|
Ether extract (% DM) |
3.3 |
3.67 |
4.03 |
6.52 |
|
Total P (% DM) |
0.72 |
0.74 |
0.77 |
0.76 |
|
Total K (% DM) |
1.22 |
1.15 |
0.95 |
0.45 |
|
Calcium (% DM) |
0.82 |
0.83 |
0.83 |
0.43 |
|
Amino acid (% DM) |
|
|
|
|
|
Aspartic acid |
0.45 |
0.56 |
0.75 |
2.87 |
|
Threonine |
0.22 |
0.3 |
0.4 |
1.39 |
|
Serine |
0.25 |
0.31 |
0.46 |
1.89 |
|
Glutamic acid |
0.85 |
1.26 |
2.13 |
4.23 |
|
Proline |
0.24 |
0.38 |
0.63 |
3.49 |
|
Glycine |
0.26 |
0.34 |
0.47 |
1.44 |
|
Alanine |
0.35 |
0.55 |
0.87 |
2.26 |
|
Valine |
0.26 |
0.36 |
0.52 |
1.62 |
|
Methionine |
0.1 |
0.13 |
0.19 |
0.85 |
|
Isoleucine |
0.21 |
0.3 |
0.42 |
1.43 |
|
Leucine |
0.42 |
0.63 |
0.96 |
3.94 |
|
Tyrosine |
0.23 |
0.29 |
0.36 |
1.53 |
|
Phenylalanine |
0.34 |
0.43 |
0.56 |
1.93 |
|
Lysine |
0.21 |
0.26 |
0.31 |
1.02 |
|
Histidine |
0.35 |
0.42 |
0.5 |
0.92 |
|
Arginine |
0.18 |
0.22 |
0.31 |
1.55 |
References:
[1]Cheng, Q.; Xu, D.; Chen, Y.; Zhu, M.; Fan, X.; Li, M.; Tang, X.; Liao, C.; Li, P.; Chen, C. Influence of Fermented-Moutai Distillers' Grain on Growth Performance, Meat Quality, and Blood Metabolites of Finishing Cattle. Front Vet Sci 2022, 9, 874453, doi:10.3389/fvets.2022.874453.
Point 2: I have a question why the authors don’t collect samples from the rumen to give more explanations? Like correlation between the end product of rumen fermentation and 16S rDNA sequencing of microbes (example: Methanogens) and rumen metabolites.
Response 2: Thank you very much for your rigorous and professional comments. Most related research efforts in ruminants to date have focused on the rumen, in large part because of its relative ease of access. The abomasum also plays an important role in nutrient digestion, while the understanding of the underlying molecular mechanisms of the abomasum under different levels of distillers grains feeding is very limited. Therefore, apart from the rumen samples, we also collected abomasum samples to study the changes in microbial community structure and metabolic profiling of Guanling cattle fed FDG diets. Indeed, samples of rumen contents were also collected for analysis, but the relevant data have been incorporated into another submitting manuscript prepared by other members of our research group, and it is not convenient to present them in this manuscript. Furthermore, we also added some comparative analyses between the abomasum and rumen samples in the Discussions section as follows: “By comparing the microbial community structure of rumen and abomasum, it was found that after feeding fermented lees, the microbial community structure of rumen and abomasum changed significantly. The main points of interaction were the significant increase of Bacteroidetes and Prevotella, and the significant decrease of Spirochetes and Tenericutes in rumen and abomasum. The difference was that Fibrobacter, Fibrobacter, Ruminococcaceae_UCG-010, Erysipelotrichosiaceae _UCG-004 in the rumen were significantly reduced; While Proteobacteria in abomasum significantly decreased, Ruminococcaceae_UCG-010 significantly increased.” [Line 443-451 marked in red]
Point 3: Figure 1 Diversity and composition of microbes in different groups. Need more resolution or change the design.
Response 3: Thank you for your comments. We have modified the overall design and the resolution of Figure 1, which is attached below. we hope that they can meet the requirements of the journal. [marked in red in Figures and Figures legends]
Figure 1 Diversity and composition of microbes in different groups. OTU comparative analysis of gastrointestinal microbiota of Guanling cattle (A) abomasum (B) cecum. Good's coverage index dilution curve from microbiota in the (C)abomasum and (D)cecum of Guanling cattle. Alpha diversity analysis of (E) abomasum and (F) cecum using the Chao1, Shannon and Simpson indices. Labeled ns indicates no significant difference. (G) Abomasal and (H) cecal principal coordinate analysis based on Bray-Curtis distance matrix. Horizontal non-metric multidimensional scale (NMDS) in the (I) abomasum and (J) cecum. BD: basal diet group; 15% FDG: a basal diet with 15% concentrate replaced by FDG; 30% FDG: a basal diet with 30% concentrate replaced by FDG.
Point 4: You need to go deeper with the scientific explanation of your findings and add more citations that support your results especially.
Response 4: Thank you very much for this comment, which is very valuable and will certainly increase the quality of our manuscript. We have followed your request for an in-depth scientific explanation of our findings in the discussion and have cited more relevant references to support my findings. The corrections were marked in red in the Discussions section of our revision. [Line 548-577 marked in red]
Point 5: I suggest make diagram to explain the correlation between feeding with fermented distillers' grains modulates gastrointestinal flora and metabolic pathways.
Response 5: Thank you very much for your comments. According to the literature review, most of the articles on gut microbiota and metabolomics have studied the correlation between differential flora and differential metabolites[2][3], while there are few studies on the correlation between microbial flora and metabolic pathways. In the original article, we used related heat maps to show the correlation between TOP20 microbial flora and TOP20 differential metabolites, but the pictures may be unclear for review. For better presentation of our results, we re-generated a correlation heatmap using Spearman correlation calculation method to analyze the correlation between the differential metabolites enriched in metabolic pathways and the TOP 15 microbiota, and we also added a comparison group of 30%FDG vs 15%FDG. [Line 393-423 marked in red].
Figure 6 Spearman level correlation between abomasum microflora and metabolites. Spearman level correlation between cecum microflora and metabolites. Correlation analysis between abomasum dominant microbials and differential metabolites in (A) 15% FDG-A VS BD-A and (B) 30% FDG-A VS BD-A and (C) 30% FDG-A VS 15% FDG-A. Correlation analysis between cecum dominant microbials and differential metabolites in (D 15%FDG-C VS BD-C and (E) 30% FDG-C VS 15% FDG-A. In the figure,Red and blue colors represent positive and negative correlations, respectively, and color gradation indicates the size of the correlation coefficient. *P < 0.05, **P < 0.01, and ***P < 0.001 indicate significant difference between the microbiota and metabolites.
References:
[2]Wang, C.; Liu, X.L.; Sun, Q.; Zhao, F.Y.; Dai, P.Q.; Li, L.X.; Hu, D.G. Apple consumption affects cecal health by regulating 12S-hydroxy-5Z,8Z,10E,14Z-eicosatetraenoic acid (12(S)-HETE) levels through modifying the microbiota in rats. Food Funct 2023, 14, 9419-9433, doi:10.1039/d3fo03207h.
[3]Ge, L.; Song, L.; Wang, L.; Li, Y.; Sun, Y.; Wang, C.; Chen, J.; Wu, G.; Pan, A.; Wu, Y., et al. Evaluating response mechanisms of soil microbiomes and metabolomes to Bt toxin additions. J Hazard Mater 2023, 448, 130904, doi:10.1016/j.jhazmat.2023.130904.
Point 6: the conclusions answer the aims of the study. To investigate the effects of partial concentrate replacement by FDG feeding in Guanling cattle.But try to focus on your findings and shorten this section and recommend the best percentage of FDG which can partially be alternative in the ruminants feeding. I think scientists have opportunities to inform future research and more research is needed.
Response 6: Thank you for your comments. We have revised and modified the Conclusions section in the revision according to your comments as follows: “The results of this study indicated that FDG replacement feeding to some extent improves the GI microbial community structure in Guanling cattle. FDG feeding increased the abundance of bacteria that promote glycolysis and degradable fiber in the GI tract of Guanling cattle, while reducing the abundance of some opportunistic pathogenic bacteria, and these changes had positive effects on the metabolism(e.g., linoleic acid metabolism, primary bile acid biosynthesis) of GI contents, thereby promoting the GI health of Guanling cattle. Comprehensive analysis of non-target metabolism showed that it was feasible to replace 15% concentrate by FDG, and these findings provided a theoretical basis for the large-scale application of FDG as a feed resource and subsequent research on beef cattle production.”[lines 581-590 marked in red]。

Reviewer 2 Report
Comments and Suggestions for Authors
Because the obtained results are greatly dependent from the ingested food, it is necessary to include:
a) the amount of offered food (was that ad libitum?)
b) how was feed intake measured? (Material and Methods)
c) the dry matter intake per treatment, and particularly, the amount of ingested FDG (Results)
Conclusions: Please delete “instead of concentrate”.
Please avoid discussion in this section: paragraph in lines 504-507 looks like discussion. Delete or rewrite.
Author Response
Response to Reviewer 2 Comments
Dear Reviewer 2,
Thank you very much for your valuable comments, and our point-to-point responses are as follows:
Point 1: the amount of offered food (was that ad libitum?)
Response 1: Thank you very much for your comments. We are very sorry for our lack of clarity. The amount of food we provide is not arbitrary, and we prepare the basic diet according to the Chinese industry standard "Beef Cattle Feeding Standard" (NY/T 815-2004), and the ratio of coarse feed to concentrate in the feed is 60:40. The specific dietary composition and nutrient level were consistent with the previous study of this subject[1], see Supplementary Table 1. Every morning before feeding, the refusals were weighed and the ration was given at the estimated intake, Dry matter intake (DMI) (kg/d) was computed daily as the difference between feed offered and the amount of refusals (DM basis). During the experiment, the animals were fed twice a day, at 9:00 and 16:00, and the cattle were allowed free access to water.
|
Supplementary Table |
||||
|
TABLE S1 | Composition and nutritional content of experimental diets varying in levels of fermented distillers’ grain (FDG) |
||||
|
Items |
BD |
15%FDG |
30%FDG |
FDG |
|
Ingredient, % DM inclusion |
||||
|
Pennisetum Sinese Roxb |
60 |
60 |
60 |
|
|
Groundcorn |
20 |
10 |
0 |
|
|
Soybean meal |
6 |
3 |
0 |
|
|
Rapeseed meal |
4 |
2 |
0 |
|
|
Wheat bran |
8 |
8 |
8 |
|
|
FDG |
0 |
15 |
30 |
|
|
Supplement |
|
|
|
|
|
Rock fine |
1.4 |
1.4 |
1.4 |
|
|
Salt |
0.05 |
0.05 |
0.05 |
|
|
Calcium bisulfate |
0.42 |
0.42 |
0.42 |
|
|
Sodium sulfate |
0.08 |
0.08 |
0.08 |
|
|
Microelement additive |
0.05 |
0.05 |
0.05 |
|
|
Total |
100 |
100 |
100 |
|
|
Chemical analysis |
|
|||
|
Dry matter (%) |
70.58 |
70.68 |
71.2 |
31.5 |
|
Gross energy (MJ/kg DM) |
15.76 |
16.6 |
16.7 |
18.69 |
|
Crude protein (% DM) |
11.4 |
12.64 |
13.88 |
21.88 |
|
Neutral detergent fiber (% DM) |
54.47 |
55.47 |
56.47 |
36.56 |
|
Acid detergent fiber (% DM) |
29.37 |
32.96 |
36.54 |
23.33 |
|
Ether extract (% DM) |
3.3 |
3.67 |
4.03 |
6.52 |
|
Total P (% DM) |
0.72 |
0.74 |
0.77 |
0.76 |
|
Total K (% DM) |
1.22 |
1.15 |
0.95 |
0.45 |
|
Calcium (% DM) |
0.82 |
0.83 |
0.83 |
0.43 |
|
Amino acid (% DM) |
|
|
|
|
|
Aspartic acid |
0.45 |
0.56 |
0.75 |
2.87 |
|
Threonine |
0.22 |
0.3 |
0.4 |
1.39 |
|
Serine |
0.25 |
0.31 |
0.46 |
1.89 |
|
Glutamic acid |
0.85 |
1.26 |
2.13 |
4.23 |
|
Proline |
0.24 |
0.38 |
0.63 |
3.49 |
|
Glycine |
0.26 |
0.34 |
0.47 |
1.44 |
|
Alanine |
0.35 |
0.55 |
0.87 |
2.26 |
|
Valine |
0.26 |
0.36 |
0.52 |
1.62 |
|
Methionine |
0.1 |
0.13 |
0.19 |
0.85 |
|
Isoleucine |
0.21 |
0.3 |
0.42 |
1.43 |
|
Leucine |
0.42 |
0.63 |
0.96 |
3.94 |
|
Tyrosine |
0.23 |
0.29 |
0.36 |
1.53 |
|
Phenylalanine |
0.34 |
0.43 |
0.56 |
1.93 |
|
Lysine |
0.21 |
0.26 |
0.31 |
1.02 |
|
Histidine |
0.35 |
0.42 |
0.5 |
0.92 |
|
Arginine |
0.18 |
0.22 |
0.31 |
1.55 |
References:
[1]Cheng, Q.; Xu, D.; Chen, Y.; Zhu, M.; Fan, X.; Li, M.; Tang, X.; Liao, C.; Li, P.; Chen, C. Influence of Fermented-Moutai Distillers' Grain on Growth Performance, Meat Quality, and Blood Metabolites of Finishing Cattle. Front Vet Sci 2022, 9, 874453, doi:10.3389/fvets.2022.874453.
Point 2: how was feed intake measured? (Material and Methods)
Response 2: Thank you for your comments. Since the data related to feed intake have been incorporated in another paper published by our group[1], it maybe not convenient to add the related methods and results for measuring feed intake to our manuscript. However, we can present the measurement of feed intake here for review. Details are as follows:
"The dry matter content of roughage was measured before feeding every day, and the feeding amount was recorded at 9:00 and 16:00 every day. At 7:30 before morning feeding, the remaining amount of the previous day was recorded and the dry matter was measured. The formula is as follows: DMI= daily feed (DM) - daily residual (DM)".
References:
[1]Cheng, Q.; Xu, D.; Chen, Y.; Zhu, M.; Fan, X.; Li, M.; Tang, X.; Liao, C.; Li, P.; Chen, C. Influence of Fermented-Moutai Distillers' Grain on Growth Performance, Meat Quality, and Blood Metabolites of Finishing Cattle. Front Vet Sci 2022, 9, 874453, doi:10.3389/fvets.2022.874453.
Point 3: the dry matter intake per treatment, and particularly, the amount of ingested FDG (Results)
Response 3: Thank you for your comments. The amount of dry matter in each experimental group is shown in the table below. Because these data have been published in articles by other members of our group[1], it is not convenient to add them to our manuscript. However, we can present it here for review.
|
Items |
BD |
15%FDG |
30%FDG |
|
Dry mattter intake(Kg/d) |
11.40±0.27 |
11.14±0.11 |
11.34±0.20 |
|
FDG intake(DM)(kg/d) |
0 |
1.71 |
3.402 |
References:
[1] Cheng, Q.; Xu, D.; Chen, Y.; Zhu, M.; Fan, X.; Li, M.; Tang, X.; Liao, C.; Li, P.; Chen, C. Influence of Fermented-Moutai Distillers' Grain on Growth Performance, Meat Quality, and Blood Metabolites of Finishing Cattle. Front Vet Sci 2022, 9, 874453, doi:10.3389/fvets.2022.874453.
Point 4: Conclusions: Please delete “instead of concentrate”.
Response 4: Thank you for your comments. We have deleted "instead of concentrate" according to your request. [line 581].
Point 5: Please avoid discussion in this section: paragraph in lines 504-507 looks like discussion. Delete or rewrite.
Response 5: Thank you for your comments. We have revised and modified the Conclusions section in the revision according to your comments as follows: “The results of this study indicated that FDG replacement feeding to some extent improves the GI microbial community structure in Guanling cattle. FDG feeding increased the abundance of bacteria that promote glycolysis and degradable fiber in the GI tract of Guanling cattle, while reducing the abundance of some opportunistic pathogenic bacteria, and these changes had positive effects on the metabolism(e.g., linoleic acid metabolism, primary bile acid biosynthesis) of GI contents, thereby promoting the GI health of Guanling cattle. Comprehensive analysis of non-target metabolism showed that it was feasible to replace 15% concentrate by FDG, and these findings provided a theoretical basis for the large-scale application of FDG as a feed resource and subsequent research on beef cattle production.”[lines 581-590 marked in red]。

Reviewer 3 Report
Comments and Suggestions for Authors
This article raises an important topic in animal science.
Writing needs to be improved.
Must be evaluated by a native speaker.
1. Introduction
Make the research hypothesis clear and add the study objective at the end of the introduction.
2. Materials and Methods
In line 118, instead of feed, replace it with roughage.
Add the composition of ingredients used in the diets to the supplementary table.
In line 125, instead of the word killed, use slaughtered.
The Student's t-test is best represented when the research has two treatments. I believe that this test choice was due to the random selection of three animals per treatment to be slaughtered. However, wouldn't there be another test that better represents your data, given that there are three treatments? Choose a test that would allow you to compare the three treatments simultaneously. This way you will also compare the 15% and 30% treatments instead of just comparing BD vs 15%FDG and BD vs 30%FDG.
3. Results
All the figures are exceedingly small. They are preventing the reader from reading the data. The figures must be improved.
Table 1: You cannot leave P-value = 0. Use P<0.001, fix it at all times that appear throughout the table.
4. Discussion
The first paragraph is too long. Reading this paragraph is tiring.
Author Response
Response to Reviewer3 Comments
Dear Reviewer 3,
Thank you very much for your valuable comments, and our point-to-point responses are as follows:
Point 1: Writing needs to be improved. Must be evaluated by a native speaker.
Response 1: Thank you very much for the constructive suggestions. We have tried our best to polish the language in the revised manuscript according to your suggestion, and we have also invited a teacher who specialize in SCI paper writing for correction of this revision. We hope that the correction will meet with approval.
- Introduction
Point 2: Make the research hypothesis clear and add the study objective at the end of the introduction.
Response 2: Thank you very much for your professional and rigorous comments. According to your request, we have revised the introduction part, clarified the research hypothesis and added the research objectives as follows: “Previous studies mainly focused on the effects of FDG on growth performance, feed conversion and digestibility of livestock and poultry. Our previous studies also showed that adding 30% FDG to the diet did not affect the growth performance, meat quality and blood metabolism of fattening cattle[15]. However, there is a lack of information on the effects of dietary fermented distiller’s grains supplementation on the gastrointestinal microbiota and its metabolites of beef cattle. Therefore, in this study, by combining 16S rDNA sequencing technology and LC-MS detection technology, we analyzed the changes of gastrointestinal microbiota and metabolites in Guanling cattle fed fermented distilling grains-based diets. These results would provide reference for the resource utilization of fermented distilling grains and the follow-up research of beef cattle production.” [lines 97-107 marked in red]
References:
[15] Cheng, Q.; Xu, D.; Chen, Y.; Zhu, M.; Fan, X.; Li, M.; Tang, X.; Liao, C.; Li, P.; Chen, C. Influence of Fermented-Moutai Distillers' Grain on Growth Performance, Meat Quality, and Blood Metabolites of Finishing Cattle. Front Vet Sci 2022, 9, 874453, doi:10.3389/fvets.2022.874453.
- 2. Materials and Methods
Point 3: In line 118, instead of feed, replace it with roughage.
Response3: Thank you for your valuable comment, We have replaced " feed" in the manuscript with "roughage". [lines 136 marked in red]
Point 4: Add the composition of ingredients used in the diets to the supplementary
table.
Response 4: Thank you very much for your comments. We have presented these data in the Supplementary material and have also cited a reference, which is previously published by our research team[1]. Since these data have already been published by other members of our group, it is not convenient to add them to our manuscript. However, we can present it here for review.
|
Supplementary Table |
||||
|
TABLE S1 | Composition and nutritional content of experimental diets varying in levels of fermented distillers’ grain (FDG) |
||||
|
Items |
BD |
15%FDG |
30%FDG |
FDG |
|
Ingredient, % DM inclusion |
||||
|
Pennisetum Sinese Roxb |
60 |
60 |
60 |
|
|
Groundcorn |
20 |
10 |
0 |
|
|
Soybean meal |
6 |
3 |
0 |
|
|
Rapeseed meal |
4 |
2 |
0 |
|
|
Wheat bran |
8 |
8 |
8 |
|
|
FDG |
0 |
15 |
30 |
|
|
Supplement |
|
|
|
|
|
Rock fine |
1.4 |
1.4 |
1.4 |
|
|
Salt |
0.05 |
0.05 |
0.05 |
|
|
Calcium bisulfate |
0.42 |
0.42 |
0.42 |
|
|
Sodium sulfate |
0.08 |
0.08 |
0.08 |
|
|
Microelement additive |
0.05 |
0.05 |
0.05 |
|
|
Total |
100 |
100 |
100 |
|
|
Chemical analysis |
|
|||
|
Dry matter (%) |
70.58 |
70.68 |
71.2 |
31.5 |
|
Gross energy (MJ/kg DM) |
15.76 |
16.6 |
16.7 |
18.69 |
|
Crude protein (% DM) |
11.4 |
12.64 |
13.88 |
21.88 |
|
Neutral detergent fiber (% DM) |
54.47 |
55.47 |
56.47 |
36.56 |
|
Acid detergent fiber (% DM) |
29.37 |
32.96 |
36.54 |
23.33 |
|
Ether extract (% DM) |
3.3 |
3.67 |
4.03 |
6.52 |
|
Total P (% DM) |
0.72 |
0.74 |
0.77 |
0.76 |
|
Total K (% DM) |
1.22 |
1.15 |
0.95 |
0.45 |
|
Calcium (% DM) |
0.82 |
0.83 |
0.83 |
0.43 |
|
Amino acid (% DM) |
|
|
|
|
|
Aspartic acid |
0.45 |
0.56 |
0.75 |
2.87 |
|
Threonine |
0.22 |
0.3 |
0.4 |
1.39 |
|
Serine |
0.25 |
0.31 |
0.46 |
1.89 |
|
Glutamic acid |
0.85 |
1.26 |
2.13 |
4.23 |
|
Proline |
0.24 |
0.38 |
0.63 |
3.49 |
|
Glycine |
0.26 |
0.34 |
0.47 |
1.44 |
|
Alanine |
0.35 |
0.55 |
0.87 |
2.26 |
|
Valine |
0.26 |
0.36 |
0.52 |
1.62 |
|
Methionine |
0.1 |
0.13 |
0.19 |
0.85 |
|
Isoleucine |
0.21 |
0.3 |
0.42 |
1.43 |
|
Leucine |
0.42 |
0.63 |
0.96 |
3.94 |
|
Tyrosine |
0.23 |
0.29 |
0.36 |
1.53 |
|
Phenylalanine |
0.34 |
0.43 |
0.56 |
1.93 |
|
Lysine |
0.21 |
0.26 |
0.31 |
1.02 |
|
Histidine |
0.35 |
0.42 |
0.5 |
0.92 |
|
Arginine |
0.18 |
0.22 |
0.31 |
1.55 |
References:
[1]Cheng, Q.; Xu, D.; Chen, Y.; Zhu, M.; Fan, X.; Li, M.; Tang, X.; Liao, C.; Li, P.; Chen, C. Influence of Fermented-Moutai Distillers' Grain on Growth Performance, Meat Quality, and Blood Metabolites of Finishing Cattle. Front Vet Sci 2022, 9, 874453, doi:10.3389/fvets.2022.874453.
Point 5: In line 125, instead of the word killed, use slaughtered.
Response 5: Thank you for your valuable comment, We have replaced " killed" in the manuscript with "slaughtered". [lines 142 marked in red]
Point 6: The Student's t-test is best represented when the research has two treatments. I believe that this test choice was due to the random selection of three animals per treatment to be slaughtered. However, wouldn't there be another test that better represents your data, given that there are three treatments? Choose a test that would allow you to compare the three treatments simultaneously. This way you will also compare the 15% and 30% treatments instead of just comparing BD vs 15%FDG and BD vs 30%FDG.
Response 6: Thank you very much for your professional comments. We have used One-way analysis of variance (ANOVA) to compare the three treatment groups, and added the data of 30%FDG vs 15%FDG comparison. For example, we added 30%FDG-A vs 15%FDG-A and 30%FDG-C vs 15%FDG-C data in OPLS-DA, volcano plot, enriched metabolic pathways, and correlation analysis, and also discussed them accordingly in the discussion. [lines 318-321、334-336、339-341、366-369、374-378 marked in red]
- Results
Point 7: All the figures are exceedingly small. They are preventing the reader from reading the data. The figures must be improved.
Response 7: Thank you very much for your comment. We've reproduced all the figures as you suggested and showed in the revision, and we hope that they can meet the requirements of the journal.
Point 8: Table 1: You cannot leave P-value = 0. Use P<0.001, fix it at all times that appear throughout the table.
Response 8: Thank you very much for your professional and rigorous comments. We apologize for our negligence. We have revised all P-values = 0 in Table 1 to P<0.001 as you suggested. Thank you again for your comments. [Table 1 marked in red]
- Discussion
Point 9: The first paragraph is too long. Reading this paragraph is tiring.
Response 9: Thank you very much for this comment. We have rewritten the first paragraph in our discussion by reducing and refining the language to make it easier and more logical to read. We hope that the correction will meet with approval. The details are as follows:”The DGs are a kind of high-quality feed resource, and their nutritional value can be improved by microbial fermentation[5]. The FDG contain a variety of beneficial microorganisms, which can improve intestinal microecology due to its nutrient decomposition and modification of intestinal microbiome, thus promoting digestion and absorption of the body [15,16], thereby improving production efficiency and ensuring animal health[17].”[lines 426-431 marked in red]
References:
[5] Iram, A.; Cekmecelioglu, D.; Demirci, A. Distillers' dried grains with solubles (DDGS) and its potential as fermentation feedstock. Appl Microbiol Biot 2020, 104, 6115-6128, doi:10.1007/s00253-020-10682-0.
[15] Wang, S.; Guo, C.; Zhou, L.; Zhong, Z.; Zhu, W.; Huang, Y.; Zhang, Z.; Gorgels, T.G.; Berendschot, T.T. Effects of dietary supplementation with epidermal growth factor-expressing Saccharomyces cerevisiae on duodenal development in weaned piglets. Brit J Nutr 2016, 115, 1509-1520, doi:10.1017/S0007114516000738.
[16] Li, Z.; Zhu, Q.; Azad, M.; Li, H.; Huang, P.; Kong, X. The Impacts of Dietary Fermented Mao-tai Lees on Growth Performance, Plasma Metabolites, and Intestinal Microbiota and Metabolites of Weaned Piglets. Front Microbiol 2021, 12, 778555, doi:10.3389/fmicb.2021.778555.
[17] O'Hara, E.; Neves, A.; Song, Y.; Guan, L.L. The Role of the Gut Microbiome in Cattle Production and Health: Driver or Passenger? Annu Rev Anim Biosci 2020, 8, 199-220, doi:10.1146/annurev-animal-021419-083952.

Reviewer 4 Report
Comments and Suggestions for Authors
Dear authors!
Thank you for sending an interesting and relevant article to Animals magazine.
In general, the article is relevant for readers, contains new information on the use of waste from the brewing industry.
However in some cases it is necessary to make corrections clarifications that will improve the text of the manuscript:
1. The sentence "With the rapid development of beef cattle industry, the shortage of feed is becoming an urgent problems, and the high cost of breeding hinders the sustainable development of animal husbandry, which makes it necessary to seek new feed ingredients." needs a reference to the literature.
2. There are very few references to literature in the literature revie.
3. In my opinion, the diet should still be given in the methodology.
4. Please cite the composition of unfermented and fermented DGs.
5. The conclusions should indicate what dosage DGs you recommend
Author Response
Response to Reviewer 4 Comments
Dear Reviewer 4,
Thank you very much for your valuable comments, and our point-to-point responses are as follows:
Point 1: The sentence "With the rapid development of beef cattle industry, the shortage of feed is becoming an urgent problem, and the high cost of breeding hinders the sustainable development of animal husbandry, which makes it necessary to seek new feed ingredients." needs a reference to the literature.
Response 1: Thank you for your comments. We have added a reference in this sentence: "With the rapid development of beef cattle industry, the shortage of feed is becoming an urgent problem, and the high cost of breeding hinders the sustainable development of animal husbandry, which makes it necessary to seek new feed ingredients[1] .” [lines 52-56 marked in red]
References:
[1] Kilama, J.; Yakir, Y.; Shaani, Y.; Adin, G.; Kaadan, S.; Wagali, P.; Sabastian, C.; Ngomuo, G.; Mabjeesh, S.J. Chemical composition, in vitro digestibility, and storability of selected agro-industrial by-products: Alternative ruminant feed ingredients in Israel. Heliyon 2023, 9, e14581, doi:10.1016/j.heliyon.2023.e14581.
Point 2: There are very few references to literature in the literature revie.
Response 2: Thank you very much for this comment, which is very valuable and will certainly make our manuscript more convincing. We cited more literature in the literature review to support our argument and make it seem more credible. Some examples are listed as follows:"In recent years, researchers have used microbial (especially probiotics) fermentation to improve the nutritional value, stability and resolve storage problems of DGs[5]." "Feeding soluble distillers' grains did not negatively affect growth performance, carcass characteristics, serum parameters, and intestinal morphology of Cherry Valley ducks[7],." "The interaction between beneficial metabolites and intestinal epithelial cells helps to maintain the intestinal mucosal barrier, thereby reducing the risk of gut-associated infections and disease[12]." etc.. [lines62-64、72-74、86-88 marked in red]
References:
[5] Iram, A.; Cekmecelioglu, D.; Demirci, A. Distillers' dried grains with solubles (DDGS) and its potential as fermentation feedstock. Appl Microbiol Biot 2020, 104, 6115-6128, doi:10.1007/s00253-020-10682-0.
[7] Zhai, S.S.; Tian, L.; Zhang, X.F.; Wang, H.; Li, M.M.; Li, X.C.; Liu, J.L.; Ye, H.; Wang, W.C.; Zhu, Y.W., et al. Effects of sources and levels of liquor distiller's grains with solubles on the growth performance, carcass characteristics, and serum parameters of Cherry Valley ducks. Poultry Sci 2020, 99, 6258-6266, doi:10.1016/j.psj.2020.07.025.
[12] Zhi, T.; Ma, A.; Liu, X.; Chen, Z.; Li, S.; Jia, Y. Dietary Supplementation of Brevibacillus laterosporus S62-9 Improves Broiler Growth and Immunity by Regulating Cecal Microbiota and Metabolites. Probiotics Antimicro 2023, doi:10.1007/s12602-023-10088-0.
Point 3: In my opinion, the diet should still be given in the methodology.
Response 3:Thank you very much for your comments. We have presented these data in the Supplementary material and have also cited a reference, which is previously published by our research team[1]. Since these data have already been published by other members of our group, it is not convenient to add them to our manuscript. However, we can present it here for review.
|
Supplementary Table |
||||
|
TABLE S1 | Composition and nutritional content of experimental diets varying in levels of fermented distillers’ grain (FDG) |
||||
|
Items |
BD |
15%FDG |
30%FDG |
FDG |
|
Ingredient, % DM inclusion |
||||
|
Pennisetum Sinese Roxb |
60 |
60 |
60 |
|
|
Groundcorn |
20 |
10 |
0 |
|
|
Soybean meal |
6 |
3 |
0 |
|
|
Rapeseed meal |
4 |
2 |
0 |
|
|
Wheat bran |
8 |
8 |
8 |
|
|
FDG |
0 |
15 |
30 |
|
|
Supplement |
|
|
|
|
|
Rock fine |
1.4 |
1.4 |
1.4 |
|
|
Salt |
0.05 |
0.05 |
0.05 |
|
|
Calcium bisulfate |
0.42 |
0.42 |
0.42 |
|
|
Sodium sulfate |
0.08 |
0.08 |
0.08 |
|
|
Microelement additive |
0.05 |
0.05 |
0.05 |
|
|
Total |
100 |
100 |
100 |
|
|
Chemical analysis |
|
|||
|
Dry matter (%) |
70.58 |
70.68 |
71.2 |
31.5 |
|
Gross energy (MJ/kg DM) |
15.76 |
16.6 |
16.7 |
18.69 |
|
Crude protein (% DM) |
11.4 |
12.64 |
13.88 |
21.88 |
|
Neutral detergent fiber (% DM) |
54.47 |
55.47 |
56.47 |
36.56 |
|
Acid detergent fiber (% DM) |
29.37 |
32.96 |
36.54 |
23.33 |
|
Ether extract (% DM) |
3.3 |
3.67 |
4.03 |
6.52 |
|
Total P (% DM) |
0.72 |
0.74 |
0.77 |
0.76 |
|
Total K (% DM) |
1.22 |
1.15 |
0.95 |
0.45 |
|
Calcium (% DM) |
0.82 |
0.83 |
0.83 |
0.43 |
|
Amino acid (% DM) |
|
|
|
|
|
Aspartic acid |
0.45 |
0.56 |
0.75 |
2.87 |
|
Threonine |
0.22 |
0.3 |
0.4 |
1.39 |
|
Serine |
0.25 |
0.31 |
0.46 |
1.89 |
|
Glutamic acid |
0.85 |
1.26 |
2.13 |
4.23 |
|
Proline |
0.24 |
0.38 |
0.63 |
3.49 |
|
Glycine |
0.26 |
0.34 |
0.47 |
1.44 |
|
Alanine |
0.35 |
0.55 |
0.87 |
2.26 |
|
Valine |
0.26 |
0.36 |
0.52 |
1.62 |
|
Methionine |
0.1 |
0.13 |
0.19 |
0.85 |
|
Isoleucine |
0.21 |
0.3 |
0.42 |
1.43 |
|
Leucine |
0.42 |
0.63 |
0.96 |
3.94 |
|
Tyrosine |
0.23 |
0.29 |
0.36 |
1.53 |
|
Phenylalanine |
0.34 |
0.43 |
0.56 |
1.93 |
|
Lysine |
0.21 |
0.26 |
0.31 |
1.02 |
|
Histidine |
0.35 |
0.42 |
0.5 |
0.92 |
|
Arginine |
0.18 |
0.22 |
0.31 |
1.55 |
References:
[1]Cheng, Q.; Xu, D.; Chen, Y.; Zhu, M.; Fan, X.; Li, M.; Tang, X.; Liao, C.; Li, P.; Chen, C. Influence of Fermented-Moutai Distillers' Grain on Growth Performance, Meat Quality, and Blood Metabolites of Finishing Cattle. Front Vet Sci 2022, 9, 874453, doi:10.3389/fvets.2022.874453.
Point 4: Please cite the composition of unfermented and fermented DGs.
Response 4: Thank you for your comments. Other members of our group have described the composition of unfermented and fermented DGs in previous published articles[1]. The details are as follows:"The main ingredients of Moutai DG are distilled sorghum and wheat that are a byproduct of the brewing processes.Moutai DG was fermented with biological starter for 15 days in a silo to generate FMDG. The ingredients of the biological starter were lactic acid bacteria, yeast, Bacillus, Bifidobacterium, Clostridium butyricum, amylase, protease, cellulase, and lipase."
References:
[1]Cheng, Q.; Xu, D.; Chen, Y.; Zhu, M.; Fan, X.; Li, M.; Tang, X.; Liao, C.; Li, P.; Chen, C. Influence of Fermented-Moutai Distillers' Grain on Growth Performance, Meat Quality, and Blood Metabolites of Finishing Cattle. Front Vet Sci 2022, 9, 874453, doi:10.3389/fvets.2022.874453.
Point 5: The conclusions should indicate what dosage DGs you recommend
Response 5: Thank you for your comments. We have revised and modified the Conclusions section in the revision according to your comments as follows: “The results of this study indicated that FDG replacement feeding to some extent improves the GI microbial community structure in Guanling cattle. FDG feeding increased the abundance of bacteria that promote glycolysis and degradable fiber in the GI tract of Guanling cattle, while reducing the abundance of some opportunistic pathogenic bacteria, and these changes had positive effects on the metabolism(e.g., linoleic acid metabolism, primary bile acid biosynthesis) of GI contents, thereby promoting the GI health of Guanling cattle. Comprehensive analysis of non-target metabolism showed that it was feasible to replace 15% concentrate by FDG, and these findings provided a theoretical basis for the large-scale application of FDG as a feed resource and subsequent research on beef cattle production". [lines581-590 marked in red]

Round 2
Reviewer 3 Report
Comments and Suggestions for Authors
The manuscript authors responded to all questions and made all suggested modifications.